# Does the Rearing Management following by Charolais Cull Cows Influence the Qualities of Carcass and Beef Meat?

**DOI:** 10.3390/foods11182889

**Published:** 2022-09-17

**Authors:** Julien Soulat, Brigitte Picard, Valérie Monteils

**Affiliations:** Université Clermont Auvergne, INRAE, VetAgro Sup, UMR Herbivores, F-63122 Saint-Genès-Champanelle, France

**Keywords:** rearing factors, rearing survey, sensory descriptors, whole life

## Abstract

This study characterized, for the first time, the rearing managements (from birth to slaughter) applied throughout the cull cows’ life and observed the effect of these managements on the carcass and meat properties. From the individual data of 371 Charolais cull cows, three rearing managements were defined and characterized with 60 rearing factors. The results showed that the rearing managements had low effects on the carcass and meat properties. For the carcass traits, only the carcass weight, and fat and longissimus (LM) colors at the level of the sixth rib were impacted. Concerning the meat, only the red color intensity, the fat aroma, the flavor intensity and persistence were affected. According to our results, this study confirmed that it is possible to produce carcass or meat with similar properties; consequently, it is difficult to favor a rearing management. However, to manage jointly both carcass and meat qualities, trade-offs are needed.

## 1. Introduction

In 2020, beef represented 20% of the meat produced in the world after the poultry and swine [1]. The European Union at 27 (**EU**) was the third producer of beef (10.2% of the world production) with France as the first producer (20.8% of the EU production, FAO, 2021). France also had the most European suckling cows (i.e., 37%), which represented the first category slaughtered in France (i.e., 28.6% of the cattle without calves) in 2021 [2,3]. In 2021, the French people consumed 22.2 kg of beef per capita, including 60% of the beef from cull cows (dairy or suckling herd) [4].

Since few years ago, the consumption trend in the EU and in France has been to reduce the quantity and to increase the quality of the beef consumed [5]. The current challenge of the beef sector was to adapt the carcass and meat productions to new market expectations (e.g., carcass weight carcass size, and meat quality) and to social expectations (e.g., animal welfare, health, and environment). The carcass and meat qualities of cull cows are sensitive to different factors (e.g., breed, stress, muscle, and aging) throughout the farm to fork continuum [6,7,8,9,10]. Many studies showed that the rearing factors could have an effect on the carcass and meat properties, in cull cows [11,12,13]. However, most studies were carried out only during the fattening period, and relatively few published results involving cull cows. In heifers, recent studies displayed the interest in considering the animal’s whole life because the periods before the fattening can have an effect on the carcass and meat qualities [14,15,16]. The aims of this study were to characterize different rearing managements (from birth to slaughter) of cull cows, and to study the effects of these rearing managements simultaneously on the carcass and meat qualities. These results could help the stakeholders of the beef sector to identify the rearing management the most interesting to product carcass and meat qualities corresponding to the target market.

## 2. Materials and Methods

### 2.1. Animals and Rearing Factors

The individual data of 371 Charolais cull cows from 39 French commercial farms in the Auvergne-Rhône-Alpes area (the second main area of sulking cattle production in France) were used in this study. The cull cows were born between December 2003 and December 2017 and slaughtered between April 2019 and December 2020.

From one face-to-face rearing survey per farm, information on the rearing management (**RM**) applied throughout the cull cows’ life were collected using questionnaires and establishing batch management practices with each farmer [17]. The cull cows’ life was divided into 4 key periods: pre-weaning period (**PWP**), growth period (**GP**), breeding period (**BP**) and fattening period (**FP**), as described in Figure 1. In addition to the rearing surveys, the composition and the nutritional values of the different concentrates used in the different diets throughout the cull cow’s life were collected from the manufacture. For the concentrates produced on the farm (e.g., wheat, corn, and barley), the nutritional values of the INRAE system were used [18]. Then, the average concentrate’s crude protein (**CP**) and net energy (**NE**) contents were calculated as described by Soulat et al. [19].

### 2.2. Slaughtering, Carcass Traits and Sampling

The cull cows were slaughtered in four French industrial slaughterhouses (SICABA, Bourbon l’Archambault; SICAREV, Roanne; SOCOPA, Villefranche-d’Allier, and Viandes de Bresse, Bourg-en-Bresse) in compliance with European regulation No 1099/2009 on the protection of animals at the time of killing [20].

The carcasses were weighted and graded for conformation and fat scores using the EUROP system [21]. Concerning the conformation score, a conversion was performed to obtain a scale (15 levels) where P− = 1 (the lowest muscular development) to E+ = 15 (the highest muscular development). After 24 h post-mortem at 2 °C, the right-hand side of the carcasses were cut at the 6th rib level and the ultimate pH were measured using a pH meter by the slaughterhouse staff in the striploin. At the cut section level, 11 other carcass traits were evaluated by trained slaughterhouse staff and described in Table 1. From a calliper, the subcutaneous fat thickness were assessed in the area presented in Figure 2. Longissimus muscle (**LM**) seepage, intermuscular fat, nerves, overall meat grain, LM meat grain, and rhomboideus meat grain (Figure 2) were evaluated using a scale from 1 to 5 with a step of 0.5 [22]. Fat and meat colors, and marbling were evaluated on LM using the reference standards [23].

On the right-hand side of the carcass, two ribs (5th and 4th ribs) localized in the chuck sale section were collected. The meat samples (2 ribs) were boneless, individually vacuum-packaged and aged 14 days at 4 °C. Then, the meat samples were frozen and stored at −20 °C until the analyses.

### 2.3. Meat Quality Evaluation

As the rib is compounded of many muscles, the meat analyses were focused on 2 muscles (LM and serratus ventralis muscle, **SV**) to characterize each meat sample. Color, texture and sensory analyses were carried out on LM, because it is the most muscle studied. Shear force analyses were carried out on SV because it is a specific muscle in the ribs of the chuck sale section [24].

The meat samples were thawed (around 48 h at 4 °C) and dissected to conserve only the LM and the SV by professional butchers (INRAE Unité Expérimentale Herbipôle, Theix, France). Each muscle sample was individually vacuum-packaged. On the dissection day, the LM samples were conserved at 4 °C before the color and sensory analyses, and the SV samples were stored at −20 °C until the shear force analyses.

#### 2.3.1. Color Assessment

Six measurements (randomly distributed) were performed on each LM sample using a spectrophotometer (Konica Minolta CR-400, Osaka, Japan) and the CIE L*a*b* system [25]. The color assessments were carried out three hours before the sensory analyses at room temperature.

#### 2.3.2. Sensory Analysis

In accordance with ISO 8586 [26], 50 persons were trained (six 1-h training sessions) to evaluate 10 sensory descriptors and one hedonic descriptor (Table 2) using a 10-cm unstructured scale (from 0 = no perception to 10 = perception very intense) and a scale anchoring (from 0 = “I don’t like at all” to 10 = “I like very much”), respectively [24]. The sensory descriptors were evaluated by sensory survey using Tastel software^®^ (ABT Informatique, Rouvroy-sur-Marne, France).

For each sensory session, 10 trained people monadically assessed 8 LM samples, using a Latin square design. Before the sensory session, 2 or 3 steaks with a 2 cm thickness were cut from each LM sample. The steaks were cooked in an aluminum foil on a plancha at 300 °C to reach an internal temperature of 55 °C. Then, steaks were cut into homogeneous pieces (size 15 × 20 × 20 mm) and kept warm.

Until the texture analyses, the rest of the LM samples were individually vacuum-packaged and frozen at −20 °C.

#### 2.3.3. Texture Profile Analysis

After thawing (around 25 min), the LM samples were cut using a cookie cutter to obtain regular cylinders (1 cm thick × 1 cm in diameter) conserved at +4 °C until the analyses. Then, each meat cylinder underwent 2 cycles of 20% compression at 4 °C using a rheometer (Kinexus pro+, Malvern Instruments, Malvern, UK) and rSpace 1.61 software (Kinexus, Malvern, UK). From the force-deformation curve, six parameters: springiness, hardness, cohesiveness, resilience, gumminess, and chewiness were calculated [27,28]. A texture profile analysis was carried out for a meat sample only when it was possible to perform the measurement on at least two meat cylinders.

#### 2.3.4. Shear Force Measurement

The SV samples were thawed at +4 °C (around 24 h), and for each sample, around 14 meat portions (between 0.9 and 1.1 cm thick × 1 cm wide) were cut in parallel to the fibers. On each raw meat portion, two shear force measurements (cut perpendicular to fibers) were carried out, using the Warner-Braztler method (Instron 5944, Elancourt, France) and Bluehill 2 software (Instron, Elancourt, France).

### 2.4. Statistical Analyses

Statistical analyses were carried out using R 4.0.5 software [29].

A descriptive analysis of the rearing factors was carried out using graphic distribution and quantile-quantile plots. As described by Soulat et al. (2018b), some quantitative rearing factors were transformed into qualitative rearing factors, according to their distribution to allow the statistical analyses.

The RM applied throughout the cull cows’ life were defined from the 60 rearing factors using the factor analysis for mixed data (**FAMD**) followed by a hierarchical clustering based on the principal components (**HCPC**). The HCPC’s dendrogram was used to determine the number of RM considered in this study. Then, the RM were characterized from ANOVA and Khi2 performed on all quantitative and qualitative rearing factors.

The effect of the RM on the carcass and meat traits was evaluated by ANOVA. For the carcass traits, the slaughterhouse and operator effects were tested in the ANOVA. If these effects were significant, a new ANOVA (mixed model) was performed, considering these effects as random effects. If these effects were not significant, they were removed, and a new ANOVA was carried out. For the sensory descriptors, mixed models considering the panelist effect as a random effect were performed. A post-analysis was realized when the effect of the RM effect was significant, using the Tukey’s test.

The “FactoMineR” package was used to realize the FAMD and the HCPC [30]. The ANOVAs were performed using the “agricolae” package [31] and the mixed models followed by Tukey’s tests using the “emmeans”, “multcompView”, and “multcomp” packages [32,33,34,35].

## 3. Results and Discussion

### 3.1. Characterisation of the Rearing Managements

From all rearing factors, the FAMD and HCPC analyses allowed to define three RM applied throughout the cull cows’ life (Table 3, Table 4, Table 5 and Table 6). These RM are summarized in the Figure 3 considering only the rearing factors discriminating the most each RM.

The first rearing management (RM-1) was followed by 149 cull cows. This management had the highest percentage of the calves from artificial insemination (Table 3). The duration of the concentrate distribution was the shortest in housing and the whole PWP. Then, 96% of the calves did not receive concentrate at pasture. During PWP, the average concentrate had the lowest CP and NE values, and the majority of the calves received forage in housing. During GP, the heifers had the shortest period in housing and the longest period outside (Table 4). The duration of the concentrate distribution was intermediate during the housing period of GP and the whole GP. During the outside period, the duration of the concentrate distribution was above 100 days for the majority of the heifers (51.0%), and 34.9% and 25.5% of the heifers ingested between 100 and 200 kg and above 200 kg of concentrate, respectively. During GP, the concentrate quantity ingested and the average concentrate’s CP value were intermediate compared to both other RM. During the outside period of GP, 34.9% and 34.2% of the heifers received an average concentrate with below 16% and between 16% and 18% of CP, respectively. The majority of the heifers (45.6%) received an average concentrate between 1.8 and 2 Mcal of NE. During the outside period of GP, the majority of the heifers (61.1%) were supplemented mainly by hay (above 80%). During the housing period of GP, 32.9% and 26.2% of the heifers ingested between 100 and 200 kg and between 200 and 400 kg of concentrate, respectively. The average concentrate had intermediate CP and NE values. During the housing period of GP, the heifers received the lowest percentage of grass silage in their diet. During BP, the duration of the concentrates distribution was intermediate and the cows ingested an intermediate concentrate quantity (Table 5). In the housing diet, 43.6%and 77.2% of the cows did not receive grass silage and corn silage, respectively. The majority of the cows (44.3%) received an average concentrate that had below 15% of CP and above 2 Mcal of NE. During FP, the cull cows had the shortest housing duration and ingested the lowest concentrate quantity (Table 6). The majority of the cull cows (59.0%) were fattened at pasture, and 36.2% of the cull cows had only the grass as a fiber source in their diet. The fattening duration at pasture was below 100 days and above 100 days for 44.3% and 28.9% of the cull cows, respectively.

The second rearing management (RM-2) was followed by 82 cull cows. In RM-2, the calves had the lowest pasture duration (Table 3). Throughout the PWP, the duration of the concentrate distribution was intermediate, and 65.8% of the calves received concentrate at pasture. At pasture, the average concentrate had below 18% of CP and above 1.8 Mcal of NE for 53.7% and 40.2% of the calves, respectively. In housing, the majority of the calves (63.4%) received forages in their diet. The heifers had the shortest GP duration, with an intermediate duration in housing and the shortest period duration at pasture (Table 4). The duration of the concentrate distribution was the shortest and 63.4% of the heifers did not receive concentrate outside. The heifers ingested the lowest concentrate quantity. During GP, the average concentrate had the lowest CP and NE values. During BP, the duration of the concentrate distribution was the shortest and the cows ingested the lowest concentrate quantity. In fact, the majority of the cows (96.3%) did not receive concentrates (Table 5). In the housing diet, 45.1% and 39.0% of the cows received between 25% and 45% of corn silage and above 60% of grass silage, respectively. Moreover, 40.2% of the cows did not receive hay. The fattening of the cull cows was mainly (97.6%) performed in housing (Table 6). The main forage in the fattening diet was straw and wrapped haylage for 29.3% and 26.8% of the cull cows.

The third rearing management (RM-3) was followed by 140 cull cows. In RM-3, there was the highest proportion of help calving (Table 3). During PWP, the calves had the longest duration with forage supplementation outside and the majority of the calves (86.4%) did not receive forage in housing. The duration of the concentrate distribution was the longest and 87.1% of the calves received concentrate outside. The average concentrate distributed outside had above 18% of CP and below 1.8 Mcal of NE for 50.7% and 70.7% of the calves, respectively. During GP, the heifers had the longest housing period and an intermediate pasture duration (Table 4). The duration of the forage supplementation outside was the shortest, and the duration of the concentrate distribution in housing and during the whole GP were the longest. The concentrate quantity intake was between 200 and 400 kg in housing and between 100 and 200 kg outside for 42.1% and 34.3% of the heifers. Moreover, 22.1% of the heifers ingested above 800 kg in housing, and outside, 48.6% did not receive a concentrate. The average concentrate in housing had the highest CP and NE values. During BP, the duration of the concentrate distribution was the longest and the cows ingested the highest concentrate quantity (Table 5). In RM-3, most cows received concentrates and the average concentrate receiving during BP could have different MAT values. The average concentrate was between 1.8 and 2 Mcal of NE for 42.9% of the cows. In the housing diet, 45.0% of the cows received above 60% of the grass silage. However, 39.3% and 47.1% of the cows did not receive hay and corn silage, respectively. The fattening of the cull cows was mainly (80.0%) performed in housing (Table 6). The main forage in the fattening diet was straw and corn silage for 36.4% and 32.1% of the cull cows. The average concentrate had the highest CP value.

Briefly, the cull cows performing the RM-1 had the longest outside period throughout their life, and they were mainly fattened at pasture. The cull cows performing the RM-2 had the lowest pasture duration throughout their life. They ingested a low concentrate quantity before the fattening, then a high concentrate quantity until the slaughter. In the RM-2, the fattening was carried out in housing with a straw or a wrapped haylage-based based-diet. Finally, the cull cows performing the RM-3 ingested the highest concentrate quantity throughout their life and were fattened in housing with a straw or a corn silage-based diet.

### 3.2. Effect of the Rearing Managements on the Carcass Traits

According to our results, the RM mainly affected the color traits of the carcass.

The carcasses had a higher ultimate pH value when the cull cows performed the RM-1 than those from both RM (Table 1). However, the differences of the ultimate pH between the three RM were weak.

At the sixth rib level, the cut section had a color more homogeneous when the cull cows performed the RM-3 than those from the RM-1. At 24h post-mortem, the LM color was darker when the cull cows performed the RM-2 than those performing the RM-1. Contrary to the RM-1, the cull cows performing the RM-2 had a shorter pasture duration and were fattening in housing with a high concentrate quantity, and those performing the RM-3 ingested the highest concentrate quantity and were fattened in housing with a straw or corn silage-based diet. According to the meat color chart, the color with a score of 4 or 5 was relatively near. In heifers, Soulat et al. [36] observed that a RM with a fattening at pasture or pasture and housing had not a significant effect on the a* parameter of the LM. These authors did not observe a significant effect of the RM on the a* parameter of the LM when the fattening was performed only in housing or at pasture and housing. For the FP, Sugimoto et al. [37] did not observe an effect of the fattening duration and the fattening diet composition on the redness of the LM, in cull cows. Contrary to our results, Priolo et al. [38] showed that cattle finished at pasture produced darker meat than those finished on the concentrate. Our results showed that the LM meat evaluated at 24 h post-mortem was not explained only by the fattening system (pasture vs. housing) in cull cows.

The carcass’s fat was yellower for the cull cows from the RM-1 compared to those from the others. These three color traits (fat color, color homogeneity, and LM color) of the carcasses were not significantly different between the RM-2 and RM-3. In the RM-1, the cull cows had the longest duration outside, and they were mainly fattened at pasture. According to Dunne et al. [39], the cattle fattened at pasture had a yellower fat than those fattened with forage and concentrate-fed. Our results were in accordance with these results. During FP, other authors did not observe effects of the fattening duration and fattening diet (e.g., concentrate quantity, and nature of forage) on the yellowness (b* parameter) of the fat color, in cull cows [7,37]. However, Holmer et al. [40] observed that the carcass fat from cull cows receiving a fattening diet with more concentrates was yellower than the carcass fat from cull cows receiving more corn silage in their diet.

The cull cows performing the RM-3 produced carcasses significantly heavier and a smoother RH meat grain compared to RM-1. These two carcass traits were not significantly different between the RM-2 and RM-3 or between the RM-1 and RM-2. Our results were in accordance with those of Soulat et al. [36], who showed that fattening at pasture produced lighter carcasses than fattening in housing or outside without grass, in the heifers. Concerning FP, many studies showed that the carcasses were heavier when the fattening duration was longer, in cull cows [37,41,42,43]. In our study, the fattening durations were not significantly different between the three RM. Franco et al. [7] did not show an effect of the fattening duration on the carcass weight. In our defined RM, the composition of the fattening diet was different (e.g., quantity of concentrates, nutritional values of the average concentrate, and nature of forage). Holmer et al. [40] and Hernandez-Calva [44] showed that the fattening diet had an effect on the carcass weight, in the cull cows. However, other studies did not observe an effect of the fattening diet composition on the carcass weight, in cull cows [37,45,46]. Moreover, Jurie et al. [8] and Fiems et al. [47] did not observe an effect of the slaughter age and the parity (after two calvings), respectively, on the carcass weight, in cull cows. As shown in the heifers, these different results showed that the carcass weight could not be explained by only one rearing factor [16]. With regard to RM, it is difficult to compare our results with previously published studies on cull cows investigating only one rearing factor of the fattening period.

To our knowledge, this is the first time that the effect of RM on the meat grain was displayed in cull cows. According to our results, Soulat et al. [24] observed an effect of the RM on the RH meat grain, in the heifers.

The carcasses from the RM-3 tended to have more intermuscular fat. This result was in accordance with Sugimoto et al. [37] and Vestergaard et al. [43], who observed that the fat rib percent was increased when the fattening duration of the cull cows was longer. However, the slaughter age [8] and the parity [47] did not affect the fat percent. The other carcass traits were not significantly different between the three RM (Table 1).

According to our results, it was difficult to favor a RM among the three RM for the carcass quality. Among the carcass traits impacted by the RM, three are related to the color. According to the target market, it will be more interesting to favor the RM-1 to have a yellower fat. The RM-2 and RM-3 allowed to produce carcasses with the same traits. This result confirmed that it is possible to produce carcasses with the same traits from different RM as showed in previous studies on heifers [24,36].

### 3.3. Effect of the Rearing Managements on the Meat Traits

In this study, due to the large number of samples collected from several slaughterhouses over a long period of time, the samples were frozen after aging until analysis. This step may have an impact on the meat properties [48]. However, as all the samples were under the same conditions, so the results were considered comparable between them.

According to our results, the flavor descriptors were the more sensitive to a modification of the RM in the cull cows, however, the observed differences were low (Table 7).

The LM meat had higher overall flavor, fat aroma, and flavor persistence when the cull cows performed the RM-3 than the RM-1. The cull cows performing the RM-3 tended to produce LM meat with more atypical flavor. Contrary to the RM-1, the cull cows performing the RM-3 ingested the highest concentrate quantity and were fattened in housing with a straw or corn silage-based diet. Contrary to Soulat et al. [13] in cull cows, an increase of the concentrate quantity intake during the fattening period induced a decrease of the LM’s flavor intensity. Hernandez-Calva et al. [44] showed a higher flavor intensity of LM meat when the cull cows received a fattening diet with hay compared to those receiving a diet with barley silage. However, a review explained that a similar flavor intensity or acceptable flavor could be obtained when the cattle were fattened at pasture or with a concentrate-based diet [49]. According to Schnell et al. [9], the fattening duration with high-energy concentrate did not affect the flavor intensity and cooked beef fat, in cull cows. Nevertheless, the fattening diet did not seem to have an effect on the atypical flavor of the LM meat, in cull cows [44,50], while the RM applied throughout the heifers’ life had an effect [15].

Concerning the tenderness and the juiciness of LM meat, in accordance with our results, many studies did not observe an effect of the slaughter age, the fattening duration, the parity, and the fattening diet, in cull cows [6,9,41,44,47,50,51]. As the cull cows are mature animals, it was more challenging to modify the physicochemical properties of LM. It was possible that LM is less sensitive to a modification of the rearing management in particular, during the fattening period. Jurie et al. [8] did not observe an effect of the cull cows’ slaughter age on the physicochemical properties of LM. This may be an element to explain that the tenderness and the juiciness were not significantly different between the RM.

According to our results, the RM did not affect the raw LM and SV meat traits (Table 7). In accordance with these results, the RM did not affect the hardness (measured by shear force and/or texture analyses) of the raw LM and SV, in the heifers [24,52]. In cull cows, Franco et al. [7] did not observe a significant effect of the fattening duration on the texture profile of cooking LM meat aging 14 days. The only tendencies were observed for the color descriptors. The LM meat from the RM-1 tended to be less light, redder, and yellower than the LM meat from both others. In accordance with our results, Gatellier et al. [53] observed that the cull cows fattened at pasture compared to those fattened with a mixed diet produced a LM meat with significant lower a* and b* values. However, these authors observed a higher L* value when cull cows were fattened at pasture compared to those fattened with a mixed diet. The fattening diet and the fattening duration did not effect the color of raw LM after aging, in cull cows [7,43,45,50].

After cooking, the LM meat was redder when the cull cows performed the RM-2 than those from the RM-3. To our knowledge, it is the first time that the effect of the RM was studied on the red color of cooking meat.

As observed for the carcass, it is difficult to favor a RM among the three for the meat quality. According to the consumer taste preferences (related to the flavor) targeted, the stakeholders of the beef chain must adapt the RM of the cull cows. If the LM meat targeted must have an intense and long persistence of flavor and a high fat aroma, the RM-3 has to be favored but it is associated with a more intense atypical flavor. According to our results, it was possible to produce LM meat with a similar quality from both RM-1 and RM-2.

With the aim to jointly manage the carcass and the meat qualities, our results did not allow to favor a RM, in the cull cows. In accordance with Soulat et al. [13], a trade-off is needed to manage these qualities in cull cows simultaneously.

## 4. Conclusions

This work studied, for the first time, the effect of the RM applied throughout the Charolais cull cows’ life on the carcass and meat qualities. Three RM were characterized from 60 rearing factors. The modification of the RM had few effects on the carcass and meat properties. Our results showed that the color (fat and lean) of the carcass and the flavor of the LM meat were the main quality traits affected by the RM. This study confirmed that it is possible to produce carcasses or LM meat with similar properties from RM with different outside durations, intake concentrate quantities or fattening systems (pasture vs. housing). In the cull cows, it was difficult to favor a RM to manage individually and jointly the carcass and meat qualities. From similar carcass and meat properties, it will be interesting to consider also, e.g., the production cost and the system’s durability, to help the beef sector stakeholders identify the RM that allowed the most ecosystem services (e.g., production, environmental, economic, and social).

## Figures and Tables

**Figure 1 foods-11-02889-f001:**
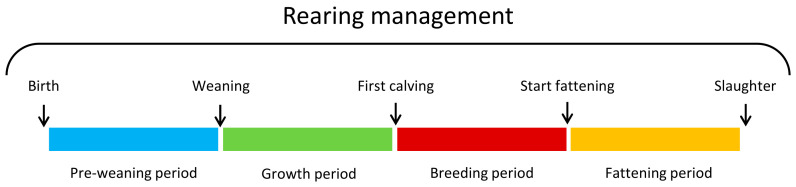
Description of the four key rearing periods during the cull cows’ whole life.

**Figure 2 foods-11-02889-f002:**
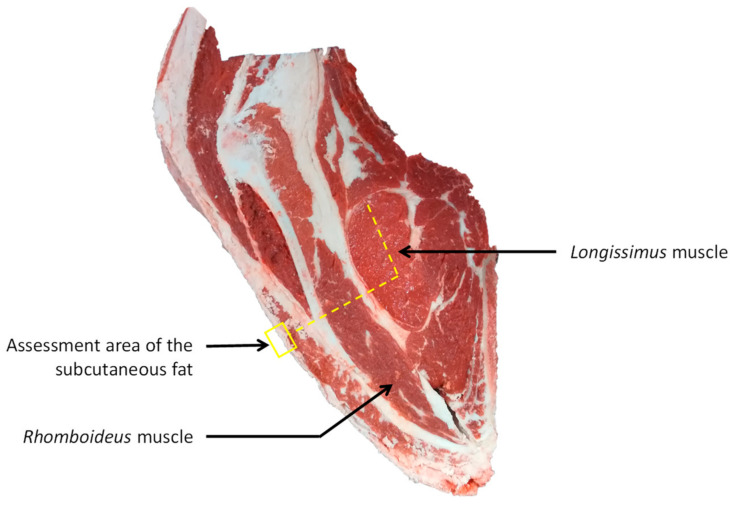
Localization of the longissimus and rhomboideus muscles on the cut section of the 6th rib and localization of the assessment area of the subcutaneous fat.

**Figure 3 foods-11-02889-f003:**
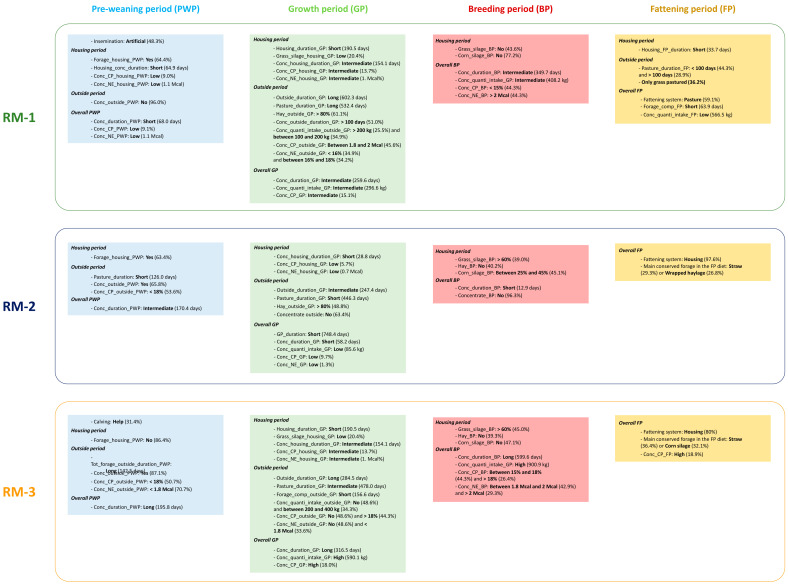
Description of the three rearing managements (RM) applied during the cull cows’ whole life, with a focus on the rearing factors characterizing the most each rearing managements and described in Table 1, Table 2, Table 3 and Table 4.

**Table 1 foods-11-02889-t001:** Effects of the three rearing managements (RM) applied during the whole cull cows’ life on carcass traits.

Carcass Traits	Description of the Carcass Trait	RM-1	RM-2	RM-3	*p*
		Emmean ± SE	Emmean ± SE	Emmean ± SE	
		*n* = 149	*n* = 82	*n* = 140	
Cold weight (kg)		450 ^b^ ± 6	460 ^ab^ ± 7	466 ^a^ ± 6	0.01
Conformation score (scale 1 to 15)	EUROP classification scales for conformation (from P- = 1 to E+ = 15)	8.7 ± 0.1	8.9 ± 0.1	8.90 ± 0.1	0.13
Fat score (scale 1 to 5)	EUROP classification scales for fat score (1 = lean to 5 = very fat)	2.9 ± 0.03	3.0 ± 0.03	3.0 ± 0.02	0.19
		** *n* ** **= 129**	** *n* ** **= 63**	** *n* ** **= 102**	
pH	The assessment of the ultimate pH was performed at 24h post-mortem	5.7 ^a^ ± 0.1	5.5 ^b^ ± 0.1	5.5 ^b^ ± 0.1	<0.001
**Assessment at the 6 th rib level**		** *n* ** **= 131**	** *n* ** **= 65**	** *n* ** **= 104**	
Longissimus muscle seepage (scale 1 to 5 scale)	Longissimus muscle seepage assessment at the 6th rib level (1 = the cut section is dry with no drop to 5 = the cut section have important drop)	2.0 ± 0.2	2.1 ± 0.2	1.9 ± 0.2	0.56
Subcutaneous fat (cm)	Measure of the subcutaneous thickness	0.8 ± 0.2	0.6 ± 0.3	1.0 ± 0.2	0.18
Intermuscular fat (1 to 5 scale)	Inter-muscular fat assessment at the 6th rib level (1 = limited development to 5 = large amount)	2.0 ± 0.2	1.9 ± 0.2	2.2 ± 0.2	0.06
Nerves (1 to 5 scale)	Nerves assessment at the 6th rib level (1 = lack of visible nerves to 5 = many visible nerves)	1.2 ± 0.1	1.2 ± 0.1	1.2 ± 0.1	0.76
Overall meat grain (1 to 5 scale)	Overall meat grain assessment at the 6th rib level (1 = smooth, soft, without harshness to 5 = very rough/granular)	2.1 ± 0.2	2.1 ± 0.2	2.1 ± 0.2	0.65
Longissimus meat grain (1 to 5 scale)	Longissimus meat grain assessment by touch at the 6th rib level (1 = smooth, soft, without harshness to 5 = very rough/granular)	1.7 ± 0.2	1.6 ± 0.3	1.6 ± 0.2	0.45
Rhomboideus meat grain (1 to 5 scale)	Rhomboideus meat grain assessment by touch at the 6th rib level (1 = smooth, soft, without harshness to 5 = very rough/granular)	1.6 ^a^ ± 0.2	1.5 ^ab^ ± 0.2	1.4 ^b^ ± 0.2	0.04
Fat color (0 to 9 scale)	Fat color assessment at the 6th rib level using the color chart described by UNECE [23]	3.2 ^a^ ± 0.3	2.6 ^b^ ± 0.3	2.8 ^b^ ± 0.3	<0.001
Color homogeneity of muscles at the 6th rib (1 to 4 scale)	Color homogeneity assessment at the 6th rib level between muscles (1 = homogeneous, 2 = bicolor, 3 = tricolor, and 4 = more than 3 colors)	1.9 ^a^ ± 0.1	1.8 ^ab^ ± 0.1	1.7 ^b^ ± 0.1	0.03
Longissimus color (0 to 7 scale)	Longissimus color assessment at the 6th rib level using the color chart described by UNECE [23]	4.2 ^b^ ± 0.3	5.0 ^a^ ± 0.3	4.6 ^ab^ ± 0.3	<0.001
Longissimus marbling (0 to 6 scale)	Longissimus marbling assessment at the 6th rib level using the marbling scale described by UNECE [23]	1.1 ± 0.2	1.2 ± 0.2	1.24 ± 0.2	0.51

Emmean = estimated marginal means. SE = standard error. *n* = number of cull cows. Values followed by different letters (a and b) are significantly different from each other at *p* ≤ 0.05.

**Table 2 foods-11-02889-t002:** Definitions of the sensory and hedonic descriptors.

Descriptors	Definition
Red color intensity	Refers to the red color intensity of the meat sample after cooking (0 = light to 10 = dark)
Initial tenderness	Facility to chew and cut the meat sample at the first bite (0 = tough to 10 = very tender)
Overall tenderness	Time and numbers of chewing required to masticate the meat sample ready for swallowing (0 = tough to 10 = very tender)
Overall juiciness	Perception of water content in the meat sample during the mastication (0 = dry to 10 = very juicy)
Presence of nerves	Quantities of nerves perceived in the meat sample (0 = none to 10 = very important)
Residue	Amount of the residue after chewing (0 = none to 10 = very important)
Flavor intensity	Global flavor intensity assessment of the beef meat (0 = none to 10 = very intense)
Fat aroma	Fat aroma intensity (0 = none to 10 = very intense)
Atypical flavor	Flavor associated with aromas that should not normally be present in meat (e.g., aftertaste, rancid) (0 = none to 10 = very intense)
Flavor persistence	Refers to remnant beef flavor duration in the mouth perceived after swallowing (0 = very quick to 10 = very long)
Overall acceptability	Overall liking (hedonic perception) of the meat sample (0 = highly disliked to 10 = highly liked)

**Table 3 foods-11-02889-t003:** Rearing factors characterizing the pre-weaning period (PWP) of the rearing managements (RM) applied during the cull cows’ whole life.

Rearing Factors	Description of the Rearing Factor	RM-1 (*n* = 149)	RM-2 (*n* = 82)	RM-3 (*n* = 140)	*p*
Quantitative Rearing Factors		Mean ± SE	Mean ± SE	Mean ± SE	
Age of the cow (years)	Age of the cow’s mother at the cow’s birth	5.5 ± 0.2	5.5 ± 0.3	4.9 ± 0.2	0.10
Housing period					
Housing_duration (days)	Numbers of days spent in stall during PWP	104.6 ± 3.4	102.8 ± 3.9	109.3 ± 3.7	0.46
Housing_conc_duration (days)	Number of days of offered concentrates in the calf diet during housing	64.9 ^b^ ± 4.5	84.6 ^a^ ± 5.1	89.3 ^a^ ± 4.4	<0.001
Conc_CP_housing_PWP (%)	Calculated average of the concentrates’ crude protein in the housing diet during PWP	9.0 ^b^ ± 0.6	15.8 ^a^ ± 0.6	14.5 ^a^ ± 0.6	<0.001
Conc_NE_housing_PWP (Mcal)	Calculated average of the concentrates’ net energy in the housing diet during PWP	1.1 ^b^ ± 0.1	1.6 ^a^ ± 0.1	1.5 ^a^ ± 0.05	<0.001
Outside period					
Pasture_duration (days)	Number of days spent at pasture during PWP	146.3 ^a^ ± 3.2	126.0 ^b^ ± 4.8	144.6 ^a^ ± 3.2	<0.001
Tot_forage_outside_duration_PWP (days)	Number of days of offered forages in the calf’s diet during outside period	96.0 ^b^ ± 6.5	93.3 ^b^ ± 7.9	142.5 ^a^ ± 3.5	<0.001
Overall PWP					
Conc_duration_PWP (days)	Number of days of offered concentrates in the calf’s diet during PWP	68.0 ^c^ ± 4.9	170.4 ^b^ ± 7.0	195.8 ^a^ ± 7.0	<0.001
Conc_CP_PWP (%)	Calculated average of the concentrates’ crude protein in the diet during PWP	9.1 ^b^ ± 0.6	15.8 ^a^ ± 0.4	17.0 ^a^ ± 0.4	<0.001
Conc_NE_PWP (Mcal)	Calculated average of the concentrates’ net energy in the diet during PWP	1.1 ^b^ ± 0.1	1.7 ^a^ ± 0.04	1.7 ^a^ ± 0.03	<0.001
**Qualitative Rearing Factors**					
Insemination type					
Artificial	Artificial insemination using frozen semen	48.3%	23.2%	20.0%	<0.001
Natural	Insemination performed by a bull	51.7%	76.8%	80.0%	
Calving					
Easy	Natural calving	16.8%	17.1%	31.4%	0.005
Help	Farmer intervention during the calving	83.2%	82.9%	68.6%	
Housing period					
Forage_housing_PWP					
No	No offered forages in calf’s diet during housing period	35.6%	36.6%	86.4%	<0.001
Yes	Offered forages in calf’s diet during housing period	64.4%	63.4%	13.6%	
Outside period					
Conc_outside_PWP					
No	No offered concentrates in calf’s diet during outside period	96.0%	34.2%	12.9%	<0.001
Yes	Offered concentrates in calf’s diet during outside period	4.0%	65.8%	87.1%	
Conc_CP_outside_PWP (Mcal)					
No	No offered concentrates in calf’s diet during outside period	96.0%	34.1%	12.9%	<0.001
≤18%	During the outside period, the calculated average of concentrate’s crude protein content was below 18%	4.0%	53.7%	50.7%	
>18%	During the outside period, the calculated average of concentrate’s crude protein content was above 18%	0%	12.2%	36.4%	
Conc_NE_outside_PWP (Mcal)					
No	No offered concentrates in calf’s diet during outside period	96.0%	34.1%	12.9%	<0.001
≤1.8 Mcal	During the outside period, the calculated average of concentrate’s net energy content was below 1.8 Mcal	0%	25.6%	70.7%	
>1.8 Mcal	During the outside period, the calculated average of concentrate’s net energy content was above 1.8 Mcal	4.0%	40.2%	16.4%	

SE = standard error. Values followed by different letters (a, b, and c) are significantly different from each other at *p* ≤ 0.05.

**Table 4 foods-11-02889-t004:** Rearing factors characterizing the growth period (GP) of the rearing managements (RM) applied during the cull cows’ whole life.

Rearing Factors	Description of the Rearing Factor	RM-1 (*n* = 149)	RM-2 (*n* = 82)	RM-3 (*n* = 140)	*p*
Quantitative Rearing Factors		Mean ± SE	Mean ± SE	Mean ± SE	
Age at the weaning (months)	Age of cow at the weaning	8.7 ± 0.08	8.7 ± 0.1	8.4 ± 0.07	0.08
Housing period					
Housing_duration_GP (days)	Number of days spent in housing during GP	190.5 ^c^ ± 6.5	247.4 ^b^ ± 10.5	284.5 ^a^ ± 5.9	<0.001
Hay_housing_GP (%)	Calculation of the hay percentage in the average housing diet across the whole GP	41.8 ^a^ ± 2.3	29.0 ^b^ ± 3.4	36.2 ^ab^ ± 2.8	0.009
Grass_silage_housing_GP (%)	Calculation of the grass silage percentage in the average housing diet across the whole GP	20.4 ^b^ ± 1.9	37.6 ^a^ ± 3.3	42.2 ^a^ ± 2.6	<0.001
Conc_housing_duration_GP (days)	Number of days of offered concentrates in the housing diet during GP	154.1 ^b^ ± 7.4	28.8 ^c^ ± 5.6	264.5 ^a^ ± 5.6	<0.001
Conc_CP_housing_GP (%)	Calculated average of concentrate’s crude protein content across the whole housing period	13.7 ^b^ ± 0.5	5.7 ^c^ ± 0.8	18.1 ^a^ ± 0.3	<0.001
Conc_NE_housing_GP (Mcal)	Calculated average of concentrate’s net energy content across the whole housing period	1.6 ^b^ ± 0.05	0.7 ^c^ ± 0.1	1.9 ^a^ ± 0.01	<0.001
Outside period					
Outside_duration_GP (days)	Number of days spent outside during GP	602.3 ^a^ ± 8.8	501.0 ^b^ ± 14.5	504.8 ^b^ ± 5.4	<0.001
Pasture_duration_GP (days)	Number of days spent at pasture during GP (cows graze)	532.4 ^a^ ± 8.6	446.3 ^c^ ± 10.1	478.0 ^b^ ± 3.9	<0.001
Forage_comp_outside_GP (days)	Number of days when forages were offered during the whole outside period of GP	231.0 ^a^ ± 13.0	243.2 ^a^ ± 20.1	156.6 ^b^ ± 9.0	<0.001
Overall GP					
GP_duration (days)	Number of days between the weaning and the first calving	792.8 ^a^ ± 7.3	748.4 ^b^ ± 13.9	789.3 ^a^ ± 3.7	<0.001
Conc_duration_GP (days)	Number of days offered concentrates in the diet during GP	259.6 ^b^ ± 6.9	58.2 ^c^ ± 8.1	316.5 ^a^ ± 4.6	<0.001
Conc_quanti_intake_GP (kg)	Total concentrate quantity intake per heifer during the whole GP	296.6 ^b^ ± 12.8	85.6 ^c^ ± 12.1	590.1 ^a^ ± 22.5	<0.001
Conc_CP_GP (%)	Calculated average of concentrate’s crude protein content across the whole GP	15.1 ^b^ ± 0.4	9.7 ^c^ ± 0.8	18.0 ^a^ ± 0.3	<0.001
Conc_NE_GP (Mcal)	Calculated average of concentrate’s net energy content across the whole GP	1.8 ^a^ ± 0.03	1.3 ^b^ ± 0.1	1.9 ^a^ ± 0.01	<0.001
**Qualitative Rearing Factors**					
Housing period					
Conc_quanti_intake_housing_GP (kg)					
0 kg	No offered concentrates during the housing period	13.4%	63.4%	0%	<0.001
<100 kg	Total concentrate quantity intake per heifer during the housing period was below 100 kg	20.1%	12.2%	1.4%	
[100 kg; 200 kg]	Total concentrate quantity intake per heifer during the housing period was between 100 kg and 200 kg	32.9%	21.9%	4.3%	
[200 kg; 400 kg]	Total concentrate quantity intake per heifer during the housing period was between 200 kg and 400 kg	26.2%	2.4%	42.1%	
[400 kg; 800 kg]	Total concentrate quantity intake per heifer during the housing period was between 400 kg and 800 kg	7.4%	0%	30.0%	
>800 kg	Total concentrate quantity intake per heifer during the housing period was above 800 kg	0%	0%	22.1%	
Outside period					
Hay_outside_GP (%)					
No	No offered hay during the outside period	11.4%	26.8%	18.6%	<0.001
<50%	In the forage supplementation during the outside period, the proportion of hay was below 50%	10.1%	2.4%	7.1%	
[50%; 80%]	In the forage supplementation during the outside period, the proportion of hay was between 50% and 80%	17.4%	21.9%	35.7%	
>80%	In the forage supplementation during the outside period, the proportion of hay was above 80%	61.1%	48.8%	38.6%	
Conc_outside_duration_GP (days)					
No	No offered concentrates during the outside period	20.1%	63.4%	48.6%	<0.001
≤100 days	During the outside period, the duration of concentrate distribution was below 100 days	28.9%	34.1%	34.3%	
>100 days	During the outside period, the duration of concentrate distribution was above 100 days	51.0%	2.4%	17.1%	
Conc_quanti_intake_outside_GP (kg)					
0 kg	No offered concentrates during the outside period	20.1%	63.4%	48.6%	<0.001
<100 kg	Total concentrate quantity intake per heifer during the outside period was below 100 kg	19.5%	26.8%	5.0%	
[100 kg; 200 kg]	Total concentrate quantity intake per heifer during the outside period was between 100 kg and 200 kg	34.9%	1.2%	34.3%	
>200 kg	Total concentrate quantity intake per heifer during the outside period was above 200 kg	25.5%	8.5%	12.1%	
Conc_CP_outside_GP (%)					
No	No offered concentrates during the outside period	20.1%	63.4%	48.6%	<0.001
<16%	Across the whole GP, the calculated average of concentrate’s crude protein content was below 16%	34.9%	17.1%	1.4%	
[16%; 18%]	Across the whole GP, the calculated average of concentrate’s crude protein content was between 16% and 18%	34.2%	19.5%	5.7%	
>18%	Across the whole GP, the calculated average of concentrate’s crude protein content was above 18%	10.7%	0%	44.3%	
Conc_NE_outside_GP (Mcal)					
No	No offered concentrates during the outside period	20.1%	63.4%	48.6%	<0.001
<1.8 Mcal	Across the whole GP, the calculated average of concentrate’s net energy content was below 1.8 Mcal	22.8%	12.2%	33.6%	
[1.8 Mcal; 2 Mcal]	Across the whole GP, the calculated average of concentrate’s net energy content was between 1.8 Mcal and 2 Mcal	45.6%	7.3%	15.0%	
>2 Mcal	Across the whole GP, the calculated average of concentrate’s net energy content was above 2 Mcal	11.4%	17.1%	2.9%	

SE = standard error. Values followed by different letters (a, b, c) are significantly different from each other at *p* ≤ 0.05

**Table 5 foods-11-02889-t005:** Rearing factors characterizing the breeding period (BP) of the rearing managements (RM) applied during the cull cows’ whole life.

Rearing Factors	Description of the Rearing Factor	RM-1 (*n* = 149)	RM-2 (*n* = 82)	RM-3 (*n* = 140)	*p*
Quantitative Rearing Factors		Mean ± SE	Mean ± SE	Mean ± SE	
Age_first_calving	Age of the cow at this first calving	2.9 ± 0.03	2.9 ± 0.03	3.0 ± 0.02	0.16
Housing period					
Housing_duration_BP (days)	Number of days spent in housing during BP	517.5 ^b^ ± 29.6	604.0 ^ab^ ± 51.7	623.0 ^a^ ± 29.0	0.05
Outside period					
Outside_duration_BP (days)	Number of days spent outside during GP	724.0 ± 50.6	808.9 ± 74.8	882.1 ± 50.4	0.10
Overall BP					
BP_duration (days)	Number of days between the first calving and the beginning of fattening	1241.4 ± 79.2	1412.9 ± 124.7	1505.1 ± 77.4	0.07
Conc_duration_BP (days)	Number of days offered concentrates in the diet during BP	349.7 ^b^ ± 30.4	12.9 ^c^ ± 8.5	599.6 ^a^ ± 30.3	<0.001
Conc_quanti_intake_BP (kg)	Total concentrate quantity intake per cow during the whole BP	408.2 ^b^ ± 38.3	23.3 ^c^ ± 15.3	900.9 ^a^ ± 56.2	<0.001
**Qualitative Rearing Factors**					
Housing period					
Hay_BP (%)					
No	Across the BP, the cows had not hay in the housing diet	5.4%	40.2%	39.3%	<0.001
<25%	Across the BP, the calculated average percentage of hay in the housing diet was below 25%	28.9%	21.9%	18.6%	
[25%; 50%]	Across the BP, the calculated average percentage of hay in the housing diet was between 25% and 50%	34.9%	34.1%	16.4%	
[50%; 60%]	Across the BP, the calculated average percentage of hay in the housing diet was between 50% and 60%	15.4%	0%	24.3%	
>60%	Across the BP, the calculated average percentage of hay in the housing diet was above 60%	15.4%	3.7%	1.4%	
Grass_silage_BP (%)					
No	Across the BP, the cows had not grass silage in the housing diet	43.6%	14.6%	14.3%	<0.001
<50%	Across the BP, the calculated average percentage of grass silage in the housing diet was below 50%	28.2%	32.9%	12.9%	
[50%; 60%]	Across the BP, the calculated average percentage of grass silage in the housing diet was between 50% and 60%	9.4%	13.4%	27.9%	
>60%	Across the BP, the calculated average percentage of grass silage in the housing diet was above 60%	18.8%	39.0%	45.0%	
Corn_silage_BP (%)					
No	Across the BP, the cows had not corn silage in the housing diet	77.2%	25.6%	47.1%	<0.001
<25%	Across the BP, the calculated average percentage of corn silage in the housing diet was below 25%	11.4%	29.3%	32.9%	
[25%; 45%]	Across the BP, the calculated average percentage of corn silage in the housing diet was between 25% and 45%	11.4%	45.1%	20.0%	
Outside Period					
Forage_comp_outside_BP (days)					
No	No forage supplementation during the outside period	32.2%	24.4%	9.3%	<0.001
<100 days	The duration of the forage supplementation was below 100 days, during the outside period	24.8%	6.1%	31.4%	
[100 days; 300 days]	The duration of the forage supplementation was between 100 and 300 days, during the outside period	19.5%	29.3%	27.9%	
[300 days; 600 days]	The duration of the forage supplementation was between 300 and 600 days, during the outside period	12.7%	14.6%	13.6%	
>600 days	The duration of the forage supplementation was above 600 days, during the outside period	10.7%	25.6%	17.9%	
Overall BP					
Conc_CP_BP (%)					
No	No offered concentrates during BP	29.5%	96.3%	4.3%	<0.001
<15%	Across the whole BP, the calculated average of concentrate’s crude protein content was below 15%	44.3%	0%	35.0%	
[15%; 18%]	Across the whole BP, the calculated average of concentrate’s crude protein content was between 15% and 18%	3.3%	3.7%	34.3%	
>18%	Across the whole BP, the calculated average of concentrate’s crude protein content was above 18%	22.8%	0%	26.4%	
Conc_NE_BP (Mcal)					
No	No offered concentrates during BP	29.5%	96.3%	4.3%	<0.001
<1.8 Mcal	Across the whole BP, the calculated average of concentrate’s net energy content was below 1.8 Mcal	19.5%	0%	23.6%	
[1.8 Mcal; 2 Mcal]	Across the whole BP, the calculated average of concentrate’s net energy content was between 1.8 Mcal and 2 Mcal	6.7%	3.7%	42.9%	
>2 Mcal	Across the whole BP, the calculated average of concentrate’s net energy content was above 2 Mcal	44.3%	0%	29.3%	

SE = standard error. Values followed by different letters (a, b, and c) are significantly different from each other at *p* ≤ 0.05.

**Table 6 foods-11-02889-t006:** Rearing factors characterizing the fattening period (FP) of the rearing managements (RM) applied during the cull cows’ whole life.

Rearing Factors	Description of the Rearing Factor	RM-1 (*n* = 149)	RM-2 (*n* = 82)	RM-3 (*n* = 140)	*p*
Quantitative Rearing Factors		Mean ± SE	Mean ± SE	Mean ± SE	
Age early fattening (months)	Age of the cull cow at the beginning of FP	6.3 ± 0.2	6.6 ± 0.3	7.0 ± 0.2	0.07
Slaughter age (months)	Age of the cull cow at the slaughter	6.6 ± 0.2	7.0 ± 0.3	7.4 ± 0.2	0.07
Housing period					
Housing_FP_duration (days)	Number of days spent in stall during the FP	33.7 ^b^ ± 4.1	124.5 ^a^ ± 6.4	109.9 ^a^ ± 3.7	< 0.001
Overall FP					
FP_duration (days)	Number of days between the beginning of FP and the slaughter	113.5 ± 4.8	125.8 ± 6.1	123.7 ± 4.3	0.17
Forage_comp_FP (days)	Number of days when forages were offered during the FP	63.9 ^b^ ± 5.1	124.5 ^a^ ± 6.4	123.7 ^a^ ± 4.3	<0.001
Conc_quanti_intake_FP (kg)	Total concentrate quantity intake per cull cow during the whole FP	566.5 ^b^ ± 30.5	1070.3 ^a^ ± 72.7	1013.6 ^a^ ± 36.2	<0.001
Conc_CP_FP (%)	Calculated average of concentrate’s crude protein content across the whole FP	17.4 ^b^ ± 0.2	17.0 ^b^ ± 0.5	18.9 ^a^ ± 0.3	<0.001
Conc_EN_FP (Mcal)	Calculated average of concentrate’s net energy content across the whole FP	1.9 ^a^ ± 0.01	1.8 ^b^ ± 0.04	1.9 ^ab^ ± 0.01	0.002
**Qualitative rearing factors**					
Pasture_duration_FP (days)					
No pasture	No pasture during the FP	26.8%	97.6%	80.0%	<0.001
≤100 days	During the FP, the number of days at pasture was below 100 days	44.3%	2.4%	12.9%	
>100 days	During the FP, the number of days at pasture was above 100 days	28.9%	0%	7.1%	
Pasture_duration_FP (days)					
No pasture	No pasture during the FP	26.8%	97.6%	80.0%	<0.001
≤100 days	During the FP, the number of days at pasture was below 100 days	44.3%	2.4%	12.9%	
>100 days	During the FP, the number of days at pasture was above 100 days	28.9%	0%	7.1%	
Fattening system					
Housing	The finishing was carried out in stall	26.8%	97.6%	80.0%	<0.001
Pasture	The finishing was carried out at pasture	59.1%	2.4%	0.7%	
Pasture&Housing	The finishing started at pasture and was finished in stall	14.1%	0%	19.3%	
Main conserved forage in the FP diet (%)					
No	No offered conserved forages in the FP diet	36.2%	2.4%	0%	<0.001
Hay_FP	The percentage of hay in the FP diet was above 62%	18.1%	8.5%	0%	
Grass_silage_FP	The percentage of grass silage in the FP diet was above77%	6.7%	6.1%	5.7%	
Wrapped_haylage_FP	The percentage of wrapped haylage in the FP diet was above71%	18.8%	26.8%	7.9%	
Corn_silage_FP	The percentage of corn silage in the FP diet was above 90%	0%	19.5%	32.1%	
Straw_FP	The percentage of straw in the FP diet was above 60%	15.4%	29.3%	36.4%	
Sorghum_FP	The percentage of sorghum silage in the FP diet was above 71%	0%	0%	5.0%	
Hay&Corn_silage_FP	The percentage of the sum of hay and corn silage in the FP diet was above 92%	1.3%	0%	6.4%	
Corn_silage&Wrapped_haylage_FP	The percentage of the sum of corn silage and wrapped haylage in the FP diet was above 91%	0%	0%	6.4%	
Hay&Corn_silage&Grass_silage_FP	The percentage of hay, corn silage and grass silage in the FP diet was 100%	3.4%	0%	0%	
Corn_silage&Grass_silage&Wrapped_haylage_FP	The percentage of corn silage, grass silage, and wrapped haylage in the FP diet was above 81%	0%	7.3%	0%	

SE = standard error. Values followed by different letters (a and b) are significantly different from each other at *p* ≤ 0.05.

**Table 7 foods-11-02889-t007:** Effects of the three rearing managements (RM) applied during the whole cull cows’ life on meat traits.

Meat Traits	RM-1	RM-2	RM-3	*p*
	Mean ± SE^1^	Mean ± SE	Mean ± SE	
Raw seratus ventralis muscle	** *n* ** **= 140**	** *n* ** **= 81**	** *n* ** **= 130**	
Shear force (N/cm^2^)	62.2 ± 1.7	60.2 ± 1.6	61.4 ± 1.4	0.70
Raw longissimus muscle (LM)				
Color	** *n* ** **= 143**	** *n* ** **= 81**	** *n* ** **= 130**	
L*	39.5 ± 0.2	39.8 ± 0.3	40.3 ± 0.2	0.07
a*	19.4 ± 0.3	19.0 ± 0.3	18.4 ± 0.3	0.06
b*	12.1 ± 0.1	11.7 ± 0.2	11.8 ± 0.1	0.06
TPA texture profile	** *n* ** **= 123**	** *n* ** **= 73**	** *n* ** **= 109**	
Springiness	0.5 ± 0.01	0.5 ± 0.01	0.5 ± 0.01	0.99
Hardness	1.6 ± 0.04	1.6 ± 0.05	1.7 ± 0.04	0.50
Cohesiveness	2.5 ± 0.1	2.4 ± 0.2	2.2 ± 0.1	0.32
Resilience	0.30 ± 0.01	0.3 ± 0.01	0.2 ± 0.01	0.14
Gumminess	4.1 ± 0.2	3.7 ± 0.3	3.6 ± 0.2	0.30
Chewiness	1.9 ± 0.1	1.7 ± 0.1	1.7 ± 0.1	0.29
Cooked LM				
Sensory descriptors (0–10 scale)	**Emmean ± SE**	**Emmean ± SE**	**Emmean ± SE**	
	** *n* ** **= 138**	** *n* ** **= 79**	** *n* ** **= 128**	
Red color intensity	4.1 ^ab^ ± 0.2	4.3 ^a^ ± 0.2	3.7 ^b^ ± 0.2	0.01
Initial tenderness	6.0 ± 0.1	6.1 ± 0.2	6.0 ± 0.1	0.72
Overall tenderness	5.5 ± 0.2	5.6 ± 0.2	5.5 ± 0.2	0.50
Overall juiciness	4.7 ± 0.2	4.8 ± 0.2	4.8 ± 0.2	0.58
Presence of nerves	2.2 ± 0.2	2.3 ± 0.2	2.3 ± 0.2	0.49
Residue	3.2 ± 0.2	3.1 ± 0.2	3.1 ± 0.2	0.86
Flavor intensity	5.9 ^b^ ± 0.1	5.9 ^ab^ ± 0.1	6.0 ^a^ ± 0.1	0.01
Fat aroma	3.6 ^b^ ± 0.2	3.8 ^ab^ ± 0.2	3.9 ^a^ ± 0.2	0.005
Atypical flavor	0.9 ± 0.2	0.8 ± 0.2	1.0 ± 0.2	0.07
Flavor persistence	4.9 ^b^ ± 0.2	4.9 ^b^ ± 0.2	5.1 ^a^ ± 0.2	0.001
Overall acceptability	5.4 ± 0.1	5.5 ± 0.2	5.4 ± 0.1	0.63

SE = standard error. *n* = number of cull cows. Emmean = estimated marginal means. Values followed by different letters (a and b) are significantly different from each other at *p* ≤ 0.05.

## Data Availability

Data is contained within the article.

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
