# Peer review of "Does the Rearing Management following by Charolais Cull Cows Influence the Qualities of Carcass and Beef Meat?"

_foods, 2022, doi:10.3390/foods11182889_

Round 1

Reviewer 1 Report

This study is the first study to characterize the rearing managements applied throughout the cull cows’ life and observe the effect of these managements on the carcass and meat properties, which results in differences in the carcass weight, and fat and longissimus colors were impacted.  Also, the red color intensity, the fat aroma, the flavor intensity and persistence were affected. Although the scientific impact is not so high, data the authors collected are valuable for improvement of meat production. For better understanding of readers, I would say that data in Figure and tables would be better if they are described in more organized manner.

Author Response

Reviewer 1

Open Review

English language and style

( ) Extensive editing of English language and style required
( ) Moderate English changes required
(x) English language and style are fine/minor spell check required
( ) I don't feel qualified to judge about the English language and style

Yes

Can be improved

Must be improved

Not applicable

Does the introduction provide sufficient background and include all relevant references?

(x)

( )

( )

( )

Are all the cited references relevant to the research?

(x)

( )

( )

( )

Is the research design appropriate?

( )

(x)

( )

( )

Are the methods adequately described?

( )

(x)

( )

( )

Are the results clearly presented?

( )

(x)

( )

( )

Are the conclusions supported by the results?

(x)

( )

( )

( )

Comments and Suggestions for Authors

This study is the first study to characterize the rearing managements applied throughout the cull cows’ life and observe the effect of these managements on the carcass and meat properties, which results in differences in the carcass weight, and fat and longissimus colors were impacted.  Also, the red color intensity, the fat aroma, the flavor intensity and persistence were affected. Although the scientific impact is not so high, data the authors collected are valuable for improvement of meat production. For better understanding of readers, I would say that data in Figure and tables would be better if they are described in more organized manner.

For better understanding of readers, we added sub-sections in the tables. As the tables were long, we divided them to avoid having the rearing factor modalities on 2 sheets. Also to help the readers, we added the header table on each table. Moreover, we modified the figure 1, for ease of understanding, by adding the same sub-sections as used in the tables and by adding color to highlight the cull cow’s live periods considered in this study.

Reviewer 2 Report

Does the Rearing Management Following by Charolais Cull Cows Influence the Qualities of Carcass and Beef Meat?

It is an interesting work dealing with a relevant meat production subject. It makes an approach that relates carcass and meat quality with three production systems for Charolais cull cows. It is always very challenging to address these issues, given the many factors influencing carcass and meat quality traits.

The article is well organized and well written and, as such, is simple to follow. The text approaches the subject scientifically and is supported by accurate references. The Tables and Figures are important for understanding the article, but some corrections are necessary to improve its clarity. Nevertheless, authors should pay attention to tables to make them easier to read. The material and methods are clearly described, allowing a perfect understanding of what has been done. However, authors should consider the possibility of better identifying the abbreviations used. The results are well presented, and the discussion is adequate. Finally, the results corroborate the conclusions.

Some detailed comments are below:

Line 34 factors can have an effect “change with” factors could have an effect

37 the interest to consider the “change with” the interest in considering the

42 rearing managements “change with”  rearing management

94 The day of the dissection “change with” On the dissection day,

133 From graphic distribution and quantile-quantile plots, a descriptive analysis of the rearing factors was carried out. “change with” A descriptive analysis of the rearing factors was carried out using graphic distribution and quantile-quantile plots.

140 characterised, 147 realized. Both North American and British spellings. Please check the text and choose one variant. Be consistent.

144 was performed considering “change with” was performed, considering

161 please check “4.ssed”.  

177 (61.1%) were supplement “change with” (61.1%) were supplemented

247-248 In RM-3, the majority of the cows received concentrates and the average concentrate receiving during BP “change with” In RM-3, most cows received concentrates, and the average concentrate received during BP

264 According to our results, the RM had mainly an effect on the color traits of the carcass. “change with” According to our results, the RM mainly affected the color traits of the carcass.

275 observe significant effect “change with” observe a significant effect

279 produced a darker meat than “change with” produced darker meat than

283 These 3 color traits “change with” These three color traits

307 parity (after 2 calving), “change with” parity (after two calvings),

313 In accordance with our “change with” According to our

316 Vestergaard et al. [43]which “change with” Vestergaard et al. [43], which

318 the parity [47] had not effect on the fat percent “change with” the parity [47] did not affect the fat percentage

328 cows, however the “change with” cows, however, the

340 with high-energy concentrate had not effect on the flavor “change with” with high-energy concentrate did not affect the flavor

351 ,50].As the cull cows are mature animals, it was possible it was more difficult to modify the physicochemical properties of LM. “change with”,50]. As the cull cows are mature animals, it was more challenging to modify the physicochemical properties of LM.

370 cull cows performing “change with” cull cows performed

380-381 In accordance with Soulat et al. [13], trade-off is needed to manage simultaneously these both qualities, in the cull cows “change with” According to Soulat et al. [13], a trade-off is needed to manage these qualities in the cull cows simultaneously.

391-393 From similar carcass and meat properties, it will be interesting to consider also e.g. the production cost, the durability of system to help the stakeholders of the beef sector to identify the RM allowed the most ecosystem services (e.g. production, environmental, economic, and social). “change with” From similar carcass and meat properties, it will be interesting to consider also, e.g. the production cost and the system's durability, to help the beef sector stakeholders identify the RM that allowed the most ecosystem services (e.g. production, environmental, economic, and social).

Author Response

Reviewer 2

Open Review

English language and style

( ) Extensive editing of English language and style required
(x) Moderate English changes required
( ) English language and style are fine/minor spell check required
( ) I don't feel qualified to judge about the English language and style

Yes

Can be improved

Must be improved

Not applicable

Does the introduction provide sufficient background and include all relevant references?

(x)

( )

( )

( )

Are all the cited references relevant to the research?

(x)

( )

( )

( )

Is the research design appropriate?

(x)

( )

( )

( )

Are the methods adequately described?

(x)

( )

( )

( )

Are the results clearly presented?

(x)

( )

( )

( )

Are the conclusions supported by the results?

(x)

( )

( )

( )

Comments and Suggestions for Authors

Does the Rearing Management Following by Charolais Cull Cows Influence the Qualities of Carcass and Beef Meat?

It is an interesting work dealing with a relevant meat production subject. It makes an approach that relates carcass and meat quality with three production systems for Charolais cull cows. It is always very challenging to address these issues, given the many factors influencing carcass and meat quality traits.

The article is well organized and well written and, as such, is simple to follow. The text approaches the subject scientifically and is supported by accurate references. The Tables and Figures are important for understanding the article, but some corrections are necessary to improve its clarity. Nevertheless, authors should pay attention to tables to make them easier to read. The material and methods are clearly described, allowing a perfect understanding of what has been done. However, authors should consider the possibility of better identifying the abbreviations used. The results are well presented, and the discussion is adequate. Finally, the results corroborate the conclusions.

For a better identifying of the abbreviation used, we applied the abbreviation in bold when it appears the first time.

Some detailed comments are below:

Line 34 factors can have an effect “change with” factors could have an effect

We have modified in the text (L 34).

37 the interest to consider the “change with” the interest in considering the

We have modified in the text (L 37)

42 rearing managements “change with”  rearing management

We have modified in the text (L 42)

94 The day of the dissection “change with” On the dissection day,

We have modified in the text (L 95-96)

133 From graphic distribution and quantile-quantile plots, a descriptive analysis of the rearing factors was carried out. “change with” A descriptive analysis of the rearing factors was carried out using graphic distribution and quantile-quantile plots.

We have modified in the text (L 134)

140 characterised, 147 realized. Both North American and British spellings. Please check the text and choose one variant. Be consistent.

We have homogenized in the text using North American spelling (L 141, 165-166)

144 was performed considering “change with” was performed, considering

We have modified in the text (L 145)

161 please check “4.ssed”. 

We have corrected in the text, it was a mistake (L 166)

177 (61.1%) were supplement “change with” (61.1%) were supplemented

We have modified in the text (L 183)

247-248 In RM-3, the majority of the cows received concentrates and the average concentrate receiving during BP “change with” In RM-3, most cows received concentrates, and the average concentrate received during BP

We have modified in the text (L 280)

264 According to our results, the RM had mainly an effect on the color traits of the carcass. “change with” According to our results, the RM mainly affected the color traits of the carcass.

We have modified in the text (L 297)

275 observe significant effect “change with” observe a significant effect

We have added “a” in the text (L 310)

279 produced a darker meat than “change with” produced darker meat than

We have deleted “a” in the text (L 314)

283 These 3 color traits “change with” These three color traits

We have modified in the text (L 318)

307 parity (after 2 calving), “change with” parity (after two calvings),

We have modified in the text (L 342)

313 In accordance with our “change with” According to our

We have modified in the text (L 348)

316 Vestergaard et al. [43]which “change with” Vestergaard et al. [43], which

We added the coma in the text (L 351)

318 the parity [47] had not effect on the fat percent “change with” the parity [47] did not affect the fat percentage

We have modified in the text (L 353)

328 cows, however the “change with” cows, however, the

We added the coma in the text (L 367)

340 with high-energy concentrate had not effect on the flavor “change with” with high-energy concentrate did not affect the flavor

We have modified in the text (L 379)

351 ,50].As the cull cows are mature animals, it was possible it was more difficult to modify the physicochemical properties of LM. “change with”,50]. As the cull cows are mature animals, it was more challenging to modify the physicochemical properties of LM.

We have modified in the text (L 392)

370 cull cows performing “change with” cull cows performed

We have modified in the text (L 409)

380-381 In accordance with Soulat et al. [13], trade-off is needed to manage simultaneously these both qualities, in the cull cows “change with” According to Soulat et al. [13], a trade-off is needed to manage these qualities in the cull cows simultaneously.

We have modified in the text (L 419-420)

391-393 From similar carcass and meat properties, it will be interesting to consider also e.g. the production cost, the durability of system to help the stakeholders of the beef sector to identify the RM allowed the most ecosystem services (e.g. production, environmental, economic, and social). “change with” From similar carcass and meat properties, it will be interesting to consider also, e.g. the production cost and the system's durability, to help the beef sector stakeholders identify the RM that allowed the most ecosystem services (e.g. production, environmental, economic, and social).

We have modified in the text (L 430-433)

Reviewer 3 Report

The study entitled “Does the Rearing Management Following by Charolais Cull Cows Influence the Qualities of Carcass and Beef Meat?” is a well-written article, followed by a well-chosen statistical approach, mixed design with factorial analysis. However, two main concerning confounders could affect the real impact of rearing management on carcass and meat quality characteristics. Pre-slaughter operations, including transport, may lead to stressful reactions, different postmortem acidification, and the development of DFD meat. Cull cows from your study came from different farms and were processed in 4 different slaughterhouses. The occurrence of the darker colour of LM measured by sensory analysis could be influenced by mentioned pre-slaughter stress. If it is possible, please add pH values of investigated samples to, in the easiest way, exclude this confounder. Another task is meat ageing with freezing and re-thawing periods that could diminish the insight of investigated variables on the meat quality properties. Besides the different genetic backgrounds and proteolysis types, destruction of muscle fibres caused by re-thawing periods could affect the meat texture characteristics, water holding capacity, and myoglobin stability leading to similar final results. A practical question of importance for both stakeholders and customers is whether the rearing management influence the carcass and fresh beef meat quality characteristics.

Author Response

Reviewer 3

Open Review

English language and style

( ) Extensive editing of English language and style required
( ) Moderate English changes required
( ) English language and style are fine/minor spell check required
(x) I don't feel qualified to judge about the English language and style

Yes

Can be improved

Must be improved

Not applicable

Does the introduction provide sufficient background and include all relevant references?

(x)

( )

( )

( )

Are all the cited references relevant to the research?

(x)

( )

( )

( )

Is the research design appropriate?

( )

(x)

( )

( )

Are the methods adequately described?

( )

(x)

( )

( )

Are the results clearly presented?

( )

(x)

( )

( )

Are the conclusions supported by the results?

( )

(x)

( )

( )

Comments and Suggestions for Authors

The study entitled “Does the Rearing Management Following by Charolais Cull Cows Influence the Qualities of Carcass and Beef Meat?” is a well-written article, followed by a well-chosen statistical approach, mixed design with factorial analysis. However, two main concerning confounders could affect the real impact of rearing management on carcass and meat quality characteristics. Pre-slaughter operations, including transport, may lead to stressful reactions, different postmortem acidification, and the development of DFD meat. Cull cows from your study came from different farms and were processed in 4 different slaughterhouses. The occurrence of the darker colour of LM measured by sensory analysis could be influenced by mentioned pre-slaughter stress. If it is possible, please add pH values of investigated samples to, in the easiest way, exclude this confounder.

In this study, since the device was based on commercial farms and slaughterhouse, we unfortunately do not have information on the pre-slaughter operations.

We added the pH values in the table 5 and some elements in the “Materials and Methods“ and “Results and Discussion“ sections. (L 72-73 ; 298-300).

As the ultimate pH was measured by the slaughterhouse staff (different pH meters), to study the effect of the rearing management on the pH, in the ANOVA, the slaughterhouse was considered as random effect. According to our results, although there are a significant effect of the rearing management on the ultimate pH, these differences were weak (a pH difference of 0.2). We did not think that this differnce of pH value had an important effect on the meat quality.We agree that pre-slaughter stress including transport has an effect on the meat properties. However, the stress can be different for animals from a same farm.

Another task is meat ageing with freezing and re-thawing periods that could diminish the insight of investigated variables on the meat quality properties. Besides the different genetic backgrounds and proteolysis types, destruction of muscle fibres caused by re-thawing periods could affect the meat texture characteristics, water holding capacity, and myoglobin stability leading to similar final results. A practical question of importance for both stakeholders and customers is whether the rearing management influence the carcass and fresh beef meat quality characteristics.

We agree with this comment, the freezing could have an impact on the muscle properties, but it was difficult for us to do otherwise. In fact, in this study as there are different slaughterhouses, a high number of samples and a long sampling period. For logistical and organizational concerns, the slaughterhouse staff had to freeze the meat samples after the aging period.

However, all the meat samples were in the same conditions for a given analysis. The results can therefore be compared with each other. In the discussion section, elements were added to mitigate the fact that the results were not obtained on fresh meat and that freezing may have had an effect on the meat properties (L 362-365)
